# Comparative Evaluation of an Easy Laboratory Method for the Concentration of Oocysts and Commercial DNA Isolation Kits for the Molecular Detection of *Cyclospora cayetanensis* in Silt Loam Soil Samples

**DOI:** 10.3390/microorganisms10071431

**Published:** 2022-07-15

**Authors:** Alicia Shipley, Joseph Arida, Sonia Almeria

**Affiliations:** 1Office of Applied Research and Safety Assessment (OARSA), Center for Food Safety and Applied Nutrition (CFSAN), Food and Drug Administration, 8301 Muirkirk Road, Laurel, MD 20708, USA; ashiple2@ur.rochester.edu (A.S.); joseph.arida@fda.hhs.gov (J.A.); 2Joint Institute for Food Safety and Applied Nutrition (JIFSAN), University of Maryland, College Park, MD 20742, USA

**Keywords:** *Cyclospora cayetanensis*, soil, detection, comparison methods, concentration flotation, commercial DNA isolation kits

## Abstract

*Cyclospora cayetanensis* is a protozoan parasite that causes foodborne outbreaks of diarrheal illness (cyclosporiasis) worldwide. Contact with soil may be an important mode of transmission for *C. cayetanensis* and could play a role in the contamination of foods. However, there is a scarcity of detection methods and studies for *C. cayetanensis* in soil. Traditional parasitology concentration methods can be useful for the detection of *C. cayetanensis*, as found for other protozoa parasites of similar size. The present study evaluated a concentration method using flotation in saturated sucrose solution, subsequent DNA template preparation and qPCR following the Bacteriological Analytical Manual (BAM) Chapter 19b method. The proposed flotation method was compared to three commercial DNA isolation kits (Fast DNA^TM^ 50 mL SPIN kit for soil (MP Biomedicals, Irvine, CA, USA), Quick-DNA^TM^ Fecal/Soil Microbe Midiprep kit (Zymo Research, Irvine, CA, USA) and DNeasy^®^ PowerMax^®^ Soil Kit (Qiagen, Hilden, Germany)) for the isolation and detection of DNA from experimentally seeded *C. cayetanensis* soil samples (5–10 g with 100 oocysts). Control unseeded samples were all negative in all methods. Significantly lower cycle threshold values (C_T_) were observed in the 100 oocyst *C. cayetanensis* samples processed via the flotation method than those processed with each of the commercial DNA isolation kits evaluated (*p* < 0.05), indicating higher recovery of the target DNA with flotation. All samples seeded with 100 oocysts (*n* = 5) were positive to the presence of the parasite by the flotation method, and no inhibition was observed in any of the processed samples. Linearity of detection of the flotation method was observed in samples seeded with different levels of oocysts, and the method was able to detect as few as 10 oocysts in 10 g of soil samples (limit of detection 1 oocyst/g). This comparative study showed that the concentration of oocysts in soil samples by flotation in high-density sucrose solutions is an easy, low-cost, and sensitive method that could be implemented for the detection of *C. cayetanensis* in environmental soil samples. The flotation method would be useful to identify environmental sources of *C. cayetanensis* contamination, persistence of the parasite in the soil and the role of soil in the transmission of *C. cayetanensis*.

## 1. Introduction

*Cyclospora cayetanensis* is an emerging foodborne protozoan parasite responsible for cyclosporiasis, a diarrheal illness associated with foodborne outbreaks worldwide. Humans become infected with *C. cayetanensis* mainly by consumption of fresh produce contaminated with sporulated oocysts that is often consumed with little or no washing [1,2]. In the United States, foodborne outbreaks, and sporadic cases of cyclosporiasis have been reported since the mid-1990s, mostly linked to imported fruit and vegetables, including cilantro, berries, basil, snow peas, and mixed salads [2]. Since 2018, the US has identified outbreaks associated to domestically grown produce. In 2021, there was a total of 1020 laboratory-confirmed domestically acquired cases from 36 states in the United States, with 70 of the people affected needing hospitalization [3].

Infected humans shed non-sporulated oocysts which sporulate and become infective in the environment in approximately 7 to 15 days [1], making direct person-to-person transmission unlikely. The presence and survival of oocysts, which are environmentally resistant, are essential for transmission. *Cyclospora cayetanensis* oocysts can contaminate plant crops via different pathways, including black water (wastewater from toilets) used for the irrigation or spraying of crops, contact with contaminated soil, infected food handlers, or hands that have been in contact with contaminated soil [2,4]. Contact with soil may be an important mode of transmission [2,5,6] and could play a role in the contamination of foods. In fact, soil contact has been considered a risk factor for *C. cayetanensis* infection in several studies in endemic areas, such as Peru, Guatemala, and Venezuela [5,6,7]. In epidemic areas such as the U.S., contact with soil was also a risk factor associated with an outbreak in Florida and the relationship remained significant after multivariate analysis [8]. However, there is a scarcity of publicly available detection-method studies and on prevalence studies on soil [9]. To our knowledge, there have only been three previous studies that analyzed and found *C. cayetanensis* contamination in soil [10,11,12], each using different sample sizes, DNA extraction and/or molecular detection methods. Therefore, an improved and standardized molecular method for detection of *C. cayetanensis* in soil is needed.

Detection of any pathogen in soil poses several limitations, which would include the type of soil analyzed and the presence of inhibitors for molecular detection in soil [13,14,15]. The limitations increase when working on a parasite such as *C. cayetanensis*, for which currently there are no in vitro or in vivo available methods for propagation. Low numbers of *C. cayetanensis* are expected to be present in soil samples as well as a heterogenous distribution of the parasite in soil samples; therefore, large soil sample collection and processing (several grams) would be advisable to increase the probability of detection of *C. cayetanensis* in soil samples. For those types of samples, most traditional methods for the detection of soil-transmitted parasites are based on concentration of the parasitic forms by sedimentation and/or flotation, before microscopic and/or molecular biology detection [16,17]. Since microscopic examination of *Cyclospora* oocysts lacks in sensitivity and does not allow for morphological differentiation and speciation of *Cyclospora* spp., the most sensitive and specific methods for detection exist in molecular methods. Among molecular methods, quantitative real-time PCR (qPCR) is a high-throughput tool and sensitive method, which allows for parasite quantification [18].

The selection of the DNA isolation protocol, as the first step towards standardization of molecular detection methods, may play a critical role in further applications such as qPCR for parasite detection [19,20]. A limited number of previous studies compared the efficacy of different commercially available DNA isolation kits for isolating the DNA from *C. cayetanensis,* on occasion in combination with other protozoa parasites [19,21,22] in fresh produce and clinical samples. However, this comparison needs to be appropriate for the matrix being analyzed [19], and none have been performed in soil. In addition, no previous comparative study of method for detection of *C. cayetanensis* included an initial oocyst concentration step. Recently, an optimized method based on sucrose solution flotation, followed by DNA extraction using mechanical grinding and specific qPCR was developed for detection of *Toxoplasma gondii* in soil samples [23,24]. *Cyclospora cayetanensis* has a similar oocyst size to *T. gondii*, so this method could be adapted to suit *C. cayetanensis* concentration needs.

The present study evaluated an easy, low-cost, and sensitive method, similar to that in *T. gondii*, using the flotation concentration of *C. cayetanensis* oocysts in sucrose solution. This concentration was then followed by DNA extraction using bead-beating and specific qPCR published in the Bacteriological Analytical Manual (BAM) Chapter 19b for *C. cayetanensis* detection [25]. All soil samples were experimentally contaminated with *C. cayetanensis* oocysts. The sensitivity of the flotation procedure was compared to other common commercial DNA isolation kits (Fast DNA^TM^ 50 mL SPIN kit for soil (MP Biomedicals), Quick-DNA^TM^ Fecal/Soil Microbe Midiprep kit (Zymo Research) and DNeasy^®^ PowerMax^®^ Soil Kit (Qiagen)) for the detection of *C. cayetanensis* in large amounts of soil samples by qPCR. Furthermore, the flotation procedure was evaluated for linearity and limit of detection in soil seeded samples with known numbers of *C. cayetanensis* oocysts.

## 2. Materials and Methods

### 2.1. Preparation of Cyclospora cayetanensis Oocysts

The oocysts used in the experiments were purified from individual human stool samples and stored in 2.5% potassium dichromate as described elsewhere [26,27]. The study was approved by the institutional review board of the FDA (protocol number 15-039F). The oocysts were enumerated using a hemocytometer on an Olympus BX51 microscope (Optical Elements Corporation, Dulles, VA, USA) [26,27]. These oocysts were washed, concentrated, and finally diluted in 0.85% NaCl to contain 10 oocysts/μL and 1 oocysts/μL for the seeding experiments. To allow comparability, the same preparation of oocysts was used for all seeding experiments in the soil samples.

### 2.2. Seeding of Soil Samples

For comparison reasons, the same soil (silt loam) was used for all the experiments. The soil had a pH of 7.0, low percentage of organic matter (3%), low percentage of sand (6%), high percentage of silt (73%) and a medium percentage of clay (21%). The soil had been kept frozen for several weeks until used and was autoclaved previously to the experiments to avoid bacterial and fungal growth.

One hundred oocysts were selected for the seeding experiments based on our previous studies, which consistently detected 100 oocysts in other matrices analyzed, including complex foods [28]. Six soil samples of 10 g each were weighted and aliquoted into 50 mL centrifuge tubes for each method and/or commercial DNA isolation kit (4 different methods). Of those, 5 samples were seeded with 100 *C. cayetanensis* oocysts, and an additional soil sample was processed unseeded to serve as a negative control for each method. The number of seeded replicates of samples analyzed were selected based on the FDA Guidelines for the Validation of Analytical Methods for the Detection of Microbial Pathogens in Foods and Feeds [29]. Each seeded sample was inoculated with 100 oocysts of *C. cayetanensis* by adding 10 μL of the 10 oocysts/μL preparation using a micropipet into the soil. A total of 35 samples were processed for the comparative study: 30 soil samples seeded (five processed by each method/variation of method) and five negative control unseeded soil samples (one for each method/variation of method).

### 2.3. Genomic DNA Isolation from Soil Samples by Concentration by Flotation in Dense Sucrose Solution and by Using Commercial DNA Isolation Kits

The concentration of *C. cayetanensis* oocysts by flotation in high density sucrose solutions and subsequent DNA isolation following the BAM Chapter 19b (Method 1, Table 1) was compared to three commercial DNA isolation kits (including a variation in one of the commercial kits) (Methods 2 to 4) directly from soil samples. The commercial DNA isolation kits were selected as the most used commercial kits for the isolation of parasite DNA in environmental samples able to process large amounts of soil samples (5–10 g).

#### 2.3.1. Concentration by Flotation in Dense Sucrose Solution and BAM Chapter 19b DNA Isolation Protocol

Each of the five *C. cayetanensis* seeded soil samples (10 g) and the unseeded control soil sample were dispersed by adding 40 mL of deionized water to the 50 mL tube containing the 10 g of soil sample, well mixed manually (1 min), and centrifuged. The supernatant was eliminated. The pellet was then mixed with cold sucrose solution (specific gravity (SG) 1.12) until completely mixed. Additional cold sugar solution was added until the tube was filled up, and after mixing again, the tubes were centrifuged at 2000× *g* for 20 min. Supernatant containing the top of the solution (20 mL) was collected into a new 50 mL tube and 30 mL of deionized water was added followed by centrifugation at 2000× *g* for 20 min. Then, the sediment was retained and washed with deionized water in a 15 mL tube. After centrifugation at 2000× *g* for 20 min, the pellet was transferred to a Fastprep^TM^ tube, centrifuged at 14,000× *g* for 4 min and kept at −20 °C until DNA isolation following the BAM Chapter 19b protocol using the FastDNA SPIN Kit for soil in conjunction with a FastPrep-24 Instrument (MP Biomedicals, Santa Ana, CA, USA) (Table 1). Original elution volume was 75 µL as recommended by the manufacturer. The total elution volume was then diluted ½ to get a final volume of 150 µL for comparison to the other commercial DNA isolation kits.

#### 2.3.2. Method 2: Commercial Fast DNA^TM^ 50 mL SPIN Kit for Soil (MP Biomedicals)

DNA from five samples of soil (10 g each) seeded with 100 *C. cayetanensis* oocysts and one unseeded control soil sample was extracted using the commercial DNA isolation kit Fast DNA^TM^ 50 mL SPIN kit for soil, which uses a bead-beater homogenizer in 50 mL tubes. The FastPrep^®^-24 bead beater instrument (MP Biomedicals, USA) was used with the FastPrep^®^ BigPrep Adapter for 50 mL tubes (2 × 50 mL tubes). The adapter only allows two samples to be processed simultaneously in the bead beater instrument. The bead-beating protocol used in this commercial kit, as well as in any of the other kits that included the use of a bead beater in the present study, consisted of two cycles of bead-beating for 50 mL tubes at a speed of 4 m/s for 45 s with at least a 45 s pause on ice between cycles. The manufacturer’s final elution using this kit is 5 mL. Afterwards, a clean-concentrator step procedure was included using the Zymo clean and concentrator-100 (Zymo Research) with a final elution volume of 150 µL.

#### 2.3.3. Method 3: Commercial Quick-DNA^TM^ Fecal/Soil Microbe Midiprep Kit (Zymo Research)

In this method, 2.5 g of soil samples is recommended but up to 5 g of soil can be processed. For comparison to the other kits, DNA from five samples of soil (5 g each) were seeded with 50 *C. cayetanensis* oocysts. DNA and the unseeded control soil sample was extracted using the commercial DNA isolation kit Quick-DNA^TM^ Fecal/Soil Microbe Midiprep kit, which also uses a bead-beater homogenizer in 50 mL tubes. The same bead beater instrument and protocol for bead-beating was used as in the previous method in conjunction with the FastPrep^®^ BigPrep Adapter for 50 mL tubes (2 × 50 mL tubes). This kit already includes a clean and concentrator step, and the final elution volume, as recommended, was 150 µL.

#### 2.3.4. Method 4: Commercial DNeasy^®^ PowerMax^®^ Soil Kit (Qiagen) with Two Variations (4a and 4b)

Variation 4a: DNA from five samples of soil (10 g each) seeded with 100 *C. cayetanensis* oocysts and the unseeded control soil sample was extracted using the commercial DNA isolation kit for soil (Dneasy^®^ PowerMax^®^ Soil Kit) following the manufacturer protocol, which does not use a bead beater homogenizer. The tubes include beads, which were vortexed for 10 min at the highest speed in a vortex with a vortex adaptor for 50 mL tubes (variation 4a). Alternatively, tubes can be placed on a water bath set at 65 °C and shaken at maximum speed for 30 min, but this step was not selected in the present study.

Variation 4b: A second set of five seeded samples and an unseeded control soil sample were processed, substituting vortexing in step 4 of the manufacturer’s protocol by two cycles of bead-beating at a speed of 4 m/s for 45 s with at least a 45 s pause on ice between cycles using the same bead beater instrument and protocol as in the previous methods, Methods 2 and 3, with a FastPrep^®^ BigPrep Adapter for 50 mL tubes (2 × 50 mL tubes).

In both variations, the recommended elution using this kit was 5 mL. Afterwards, a clean-concentrator step (Zymo clean and concentrator-100 (Zymo Research) was included and the final elution volume was 150 µL for comparison purposes.

### 2.4. Quantitative Real-Time BAM Chapter 19b qPCR for Soil Samples

After DNA was extracted by the different protocols/methods, qPCR was performed following BAM Chapter 19b methodology [18,25]. Briefly, molecular detection of *C. cayetanensis* was performed by a duplex reaction, targeting both the specific *C. cayetanensis* multicopy 18S ribosomal RNA gene and an exogenous internal amplification control (IAC). The IAC is designed to monitor for any matrix-associated inhibition of the reaction. The qPCR was performed on an Applied Biosystems 7500 Fast Real-Time PCR System (ThermoFisher Scientific, Waltham, MA, USA). A commercially prepared synthetic gBlocks gene fragment (Integrated DNA Technologies, Coralville, CA, USA) (HMgBlock135m) was used as a positive control for amplification of the *C. cayetanensis* 18S rRNA gene [18]. Each experimental qPCR run consisted of triplicate reactions for 45 cycles of the study samples, a non-template control (NTC), and positive controls containing 10-fold serial dilutions from 10^3^ to 10 copies of the synthetic positive control. Runs were only considered valid if all three replicates of the positive control reactions produced the expected positive result and the NTC was negative. Samples were only considered positive when one or more of the replicates produced a positive result with a cycle threshold (C_T_) ≤ 38.0 for the 18S target. According to this method, undetermined reactions would be considered inconclusive if the IAC reaction failed or produced an average C_T_ value more than three cycles higher when compared to the NTC for the same assay or if the IAC reaction failed completely [18,25].

### 2.5. Linearity and Limit of Detection of C. cayetanensis by the Flotation Protocol in Soil Samples

To analyze the linearity of detection of the flotation method (Method 1), samples of 10 g of soil were seeded with known numbers of oocysts (1000 oocysts: 4 samples; 200 oocysts: 2 samples; 100 oocysts: 5 samples; 50 oocysts: 1 sample; 20 oocysts: 10 samples; and 10 oocysts: 10 samples) and were processed individually following the flotation protocol, DNA extraction and qPCR indicated for Method 1. Positive sample average C_T_ value results at each seeded level (all samples analyzed positive up to 20 oocysts: 8 positive samples and 10 oocysts: 3 samples positive) were plotted and a linear trendline equation and the corresponding R-square value were calculated.

To assess the limit of detection of the flotation method in soil samples, a low number of oocysts (20 and 10 oocysts) were seeded in sets of ten 10 g samples following the Method 1 protocol, DNA extraction and qPCR. The corresponding C_T_ values per sample and rate of positive recovery in each seeding level were then calculated.

### 2.6. Quantitative Real-Time Mitochondrial qPCR for Soil Samples

While *C. cayetanensis* multicopy 18S ribosomal RNA gene is used in the validated Chapter 19b methodology, we have been working on a new method with a different multi-copy target in the mitochondrial genome of *C. cayetanensis* (Mit1C assay). The Mit1C target was identified in silico using BLAST searches against *C. cayetanensis* and other genera/species (e.g., *Eimeria* spp., and *Isospora* spp.) in the Apicomplexa phylum. The target is a 205 bp region with a 100% consensus to all reported *C. cayetanensis* sequences in the NCBI database, being species specific. We tested the new method with DNA from samples using Method 1 (Section 2.3.1) and 4b (Section 2.3.4 variation 4b), as representative samples with the new TaqMan real-time PCR duplex assay, which targets both the new mitochondria *C. cayetanensis* gene and the same exogenous IAC target as described in BAM Chapter 19b. The new qPCR was performed using the following primers: Mit1C-f 5′-TCTATTTTCACCATTCTTGCTCAC-3′ and Mit1C-r 5′-TGGACTTACTAGGGTGGAGTCT-3′. The TaqMan probe (Mit1C-P) was labeled with a 5′ 6-FAM fluorophore, a 3′ Iowa Black FQ quencher, and an internal ZEN quencher: FAM 5′-AGGAGATAGAATGCTGGTGTATGCACC-3′ Iowa Black^®^ FQ. The duplex real-time PCR assay was performed on the Applied Biosystems 7500 Fast Real-Time PCR System (Thermo-Fisher Scientific) in fast mode using the 5× PerfeCTa Multiplex qPCR Toughmix Low ROX as an enzyme (Quantabio Cat. No. 95149-250) with ROX reference dye included. Final concentrations of primers and probes were 0.4 and 0.25 µM, respectively, for both the *C. cayetanensis*–Mit1C specific reaction and for the IAC reaction. The concentration of the synIAC ultramer oligo target was 10^6^ copies per µL. Reactions were performed using 2.0 µL of template in a final reaction volume of 20 µL. The amplification protocol consisted of an initial step of 95 °C for 3 min followed by 40 cycles of 95 °C for 15 s and 61 °C for 60 s. Data were collected during the 61 °C step. Analysis was performed using the Applied Biosystems 7500 Software v.2.3 with a baseline setting from 6 to 15 cycles. The threshold for the Mit1C target reactions was 0.08 and for the IAC target reactions was 0.05.

### 2.7. Statistically Analysis

Positive detection rates were calculated as the percentage of inoculated samples, which gave a positive result in the samples processed by each method and/or experiment. Statistically significant differences in C_T_ values between different methods of DNA isolation by BAM Chapter 19b qPCR were analyzed by one-way analysis of variance and multiple comparisons by Tukey’s multiple comparison test. Statistically significant differences in C_T_ values between two different methods (Method 1 and 4b) by Mit1C qPCR were analyzed by non-parametric “t” Mann–Whitney test. Statistical analyses were performed using GraphPad version 9.1 (GraphPad, San Diego, CA, USA), with a *p* value of ≤0.05 indicating statistical differences. When samples were undetermined in the qPCR reaction, the negative samples were excluded from the calculation for average C_T_ values.

## 3. Results

### 3.1. Comparison of Detection of C. cayetanensis in the Different Methods/Kits

All control unseeded soil samples analyzed were negative for the presence of the parasite in any of the methods. The linearity and efficiency of the standard curves using positive synthetic controls (10^3^ to 10 copy numbers) included in each qPCR in the study showed good linearity (R ≥ 0.98) and good efficiency of reaction (close to 100%). Positive results (C_T_ values for specific *C. cayetanensis* 18S sRNA and the IAC C_T_ values for seeded samples and unseeded controls) are shown in Table 2. All five replicates of soil samples seeded with 100 *C. cayetanensis* oocysts were found to be positive for the presence of the parasite using the flotation method and the Quick-DNA^TM^ Fecal/Soil Microbe Midiprep kit (Zymo Research) (100% detection rate) (Table 2). Only one of the five replicates of soil seeded with 100 *C. cayetanensis* oocysts was found to be positive in the detection of the parasite (C_T_ value 36.2) when DNA from 10 g was extracted using the original protocol for the commercial DNeasy^®^ PowerMax^®^ Soil Kit, which includes vortexing (20% detection rate) (Method 4a). When the DNeasy^®^ PowerMax^®^ Soil Kit protocol was modified to substitute vortexing by bead-beating, four of five samples were positive to the presence of *C. cayetanensis* DNA using DNeasy^®^ PowerMax^®^ Soil Kit (80% detection rate) (Method 4b) (Table 2). Similarly, four of five samples were positive to the presence of *C. cayetanensis* DNA using the kit Fast DNA^TM^ 50 mL SPIN kit for soil (MP Biomedicals) (80% detection rate) (Table 2).

Statistically significant differences were observed in the average C_T_ values for *C. cayetanensis* 18S rRNA in samples seeded with 100 *C. cayetanensis* oocysts by qPCR among methods, with the flotation method showing statistically lower C_T_ values for *C. cayetanensis* 18S rRNA (better detection) than each of the other direct DNA isolation methods (*p* < 0.05). No statistically significant differences were observed between the Fast DNA^TM^ 50 mL SPIN kit for soil (MP Biomedicals), the Quick-DNA^TM^ Fecal/Soil Microbe Midiprep kit (Zymo Research) and DNeasy^®^ PowerMax^®^ Soil Kit (Qiagen)) using the modified protocol including bead-beating (*p* > 0.05) (Table 2, Figure 1).

No inhibition was observed based on the IAC C_T_ values [18] in the samples of soil processed by flotation or by the DNeasy^®^ PowerMax^®^ Soil Kit (Qiagen) (Table 2). One sample showed some degree of inhibition (IAC levels higher than 3 C_T_ values compared to the NTC of the same assay) using the Fast DNA^TM^ 50 mL SPIN kit for soil (MP Biomedicals) and two samples showed inhibition using the Quick-DNA^TM^ Fecal/Soil Microbe Midiprep kit (Zymo Research) (highlighted in red, Table 2). However, detection of 18S rRNA was achieved in those samples.

### 3.2. Linearity of C. cayetanensis and Limit of Detection of Method 1: Flotation Protocol in Soil Samples

The standard curve of the whole range of 10 g soil samples seeded with known numbers of oocysts (from 1000 oocysts to 10 oocysts) was linear and showed an excellent R-square value (R^2^: 0.9733) (Figure 2).

In samples seeded with low numbers of oocysts using the flotation method, detection rates were 80% and 30% for 20 oocysts seeding level and 10 oocysts seeding level, respectively (Table 3). A fractional level of detection (25–75% positive samples of the analyzed samples) was obtained in samples seeded with 10 oocysts. Therefore, as few as 1 oocyst/g of sample were detected using flotation (Method 1).

### 3.3. Quantitative Real-Time Mitochondrial qPCR for Soil Samples

Results of Method 1 and Method 4b, as representative of the two DNA isolation methods, were compared using a new Mitochondrial *C. cayetanensis* qPCR (Table 4).

All five replicates of soil samples seeded with 100 *C. cayetanensis* oocysts were found to be positive for the presence of the parasite using the flotation method (Method 1) and the DNeasy^®^ PowerMax^®^ Soil Kit modified to substitute vortexing by bead-beating (Method 4b) (100% detection rate) (Table 4). Unseeded samples were negative using Mit1C assay in Methods 1 and 4b. No inhibition was observed based on IAC values (Table 4). Statistically significant differences were observed in the average C_T_ values for *C. cayetanensis* in samples seeded with 100 *C. cayetanensis* oocysts by Mit1C qPCR between both methods. The flotation method showed statistically lower C_T_ values for *C. cayetanensis* Mit1C (better detection) (*p* < 0.05), as it was previously observed by 18S *C. cayetanensis* qPCR.

## 4. Discussion

The present study evaluated a reliable and sensitive procedure for the detection of *C. cayetanensis* in large soil samples (weighing several grams), which included a flotation concentration of oocysts in saturated sucrose solutions and subsequent DNA isolation and qPCR, as reported in the BAM Chapter 19b method (named as flotation procedure). This procedure showed several advantages compared to other commonly used commercially available kits for DNA isolation from large samples of soil (5–10 g). First, the flotation procedure provided very high detection rates (100% of the samples seeded with 100 *C. cayetanensis* oocysts analyzed in the study), with statistically significantly lower C_T_ values for 18S rRNA specific for *C. cayetanensis* by qPCR than commercially available kits. Second, no inhibition was observed in the type of soil and samples processed based on the IAC included in the qPCR assay, while some degree of inhibition was observed in two of the commercial methods. The same conclusions were reached using a new qPCR method targeting the *C. cayetanensis* mitochondria gene in samples from Methods 1 (flotation) and 4b (direct DNA extraction using a commercial soil kit). Third, after flotation, due to the small sample pellet collected, more samples could be simultaneously processed for DNA isolation in a faster fashion, and in addition the cost of processing was cheaper using a smaller column DNA isolation kit (low-cost method).

Presently, the methods used for recovery of zoonotic parasites from soil, including *C. cayetanensis*, are not standardized [24]. To our knowledge, only three studies detected *C. cayetanensis* by molecular methods in soil [10,11,12] and the results in those studies are not comparable due to the different methodologies, sample sizes and/or gene targets. One of those methods [10] processed 10 g of soil on percoll–sucrose gradients and used a commercial genomic DNA isolation kit and qPCR and melting curve analysis. No bead-beating was included in the processing of DNA extraction of the soil samples. Direct DNA extraction using commercial kits was performed in the other two studies [11,12], with small soil samples (250 mg) processed in the most recent study [12].

A simple, cost-effective, standardized, sensitive method for the detection of *C. cayetanensis* in large soil samples in the presence of soil inhibitors was needed. The protocol used in soil samples in the present study substituted the use of an overnight step in sulfuric acid to kill bacteria and fungal contaminants [23] by autoclaving the soil sample before any seeding and experimental procedures. The present method for flotation used cold sucrose solution (SG of 1.12), which has been shown to be effective in floating and isolating *T. gondii* and *Cryptosporidium* spp. oocysts from soil samples [23,24,30]. Some fastidious and tricky steps used in previous detection studies for other parasites in soil were avoided, such as thermal shock of three freeze–thaw cycles of 4 h each [11] and the use of gradients with the sucrose solutions [23,24]. The present method therefore is easier and faster than those previously described for detection of other protozoan parasites in soil [23,24].

A main challenge associated with nucleic acid isolation from parasites, from helminthic eggs to protozoan oocysts, is their thick and tough protective exterior wall [31,32]. Protozoa oocyst walls present complex physiochemical features, and conventional DNA isolation systems provide poor performances in extracting parasite DNA from different matrices, such as stool [31]. Sporulated *Cyclospora* oocysts possess a double wall, which would limit ease of breaking the oocyst wall before DNA isolation. Although several freeze–thaw cycles can be used for the DNA isolation of protozoan parasites, including *C. cayetanensis* [23,33], the use of bead-beating instruments is becoming the most used method for the break-up of walls in the DNA isolation of *C. cayetanensis* oocysts in other matrices, such as produce [11,18,19,26,27,28], and it was included in the manufacturer’s protocols in three of the four methods compared in the present study. In fact, in the only commercial kit that did not include bead-beating in the manufacturer’s instructions (DNeasy PowerSoil kit), which used vortexing instead, DNA isolation was poor (20% recovery rate in the samples analyzed). Vortexing might not have been enough to break the oocyst wall and extract DNA from most oocysts of *C. cayetanensis*. In a previous study [19], the DNeasy PowerSoil kit protocol was also modified using a bead-beater instead of vortexing, and this kit performed better than the UNEX-based DNA isolation in the detection of *C. cayetanensis* in berries [19]. In the present study, when the use of bead-beating instead of vortexing for the DNeasy PowerSoil kit was assessed, a better recovery rate (up to 80%) was observed and the obtained C_T_ values for specific *C. cayetanensis* 18S sRNA were similar to the other commercial kits analyzed, which used bead-beating. Previous studies in other parasites showed similar conclusions. Mechanical pretreatment was found to be necessary to improve DNA isolation for *Cryptosporidium* spp. Oocysts in stools [31], and DNA isolation of eggs of the soil-transmitted helminth *Ascaris* using bead-beating techniques produced a higher yield of DNA compared to a kit based solely on enzymatic reactions [20].

Another positive aspect of the initial concentration of *C. cayetanensis* oocysts by flotation in high density sucrose solution was the lack of inhibition in the soil samples processed in the present study (low percentage of sand (6%), high percentage of silt (73%) and a medium percentage of clay (21%)). The presence of inhibitors can clearly be an issue to consider when selecting the best DNA isolation approach, and different matrices are likely to have different inhibitors in varying quantities [13,14,15,19]. As indicated, the BAM Chapter 19b qPCR includes an internal control (IAC) to investigate the presence of any inhibition and false negative samples due to the presence of inhibitors, and we did not find inhibition of the qPCR reaction using the flotation concentration method. Some inhibition was observed in two of the other commercial methods (one sample showed inhibition using Fast DNA^TM^ 50 mL SPIN kit for soil and two samples showed inhibition using the Quick-DNA^TM^ Fecal/Soil Microbe Midiprep kit). Previous studies have shown that inhibition is not a common feature in the molecular detection of *C. cayetanensis* following BAM Chapter 19b in produce [18,26,34,35], but if inhibition occurs, commercially available clean-up DNA methods have been found to be useful to avoid inhibition for detection of *C. cayetanensis* in cilantro samples with soil [36]. Further studies will need to be performed to confirm the lack of inhibition using the present protocol in different types of soil since different soils are likely to have different inhibitors in varying quantities.

Importantly, our results show that the flotation procedure was the most sensitive method for the detection of *C. cayetanensis* in soil. Statistically significant lower C_T_ values (better detection) for *C. cayetanensis* 18S rRNA and Mit1C were observed in the flotation method compared to other direct DNA isolation methods in the soil samples analyzed. In addition, high detection rates (100% in samples seeded with 100 *C. cayetanensis* oocysts analyzed in the study) were observed using the flotation step, while detection rates were lower using the other DNA isolation methods, with the exception of the Quick-DNA^TM^ Fecal/Soil Microbe Midiprep kit, which showed the same detection rate as the flotation procedure by 18S qPCR. The present method showed linearity of detection in the whole range of samples seeded with different oocysts numbers (from 1000 to 10 oocysts) and showed a limit of detection of 1 oocyst/g of soil since as few as 10 oocysts were detected in 10 g of soil. In addition, it allowed the simultaneous analysis of a larger number of samples in the processing of several grams of soil samples. Previous studies of detection of *C. cayetanensis* in soil did not evaluate the limit of detection of their methods in soil [10,11,12]. In produce, 20 oocysts of *C. cayetanensis* were detected in berries using a DNeasy PowerSoil kit with a bead-beating step [19]. Using the BAM Chapter 19b method in produce, as few as five oocysts have been detected in cilantro and raspberries [18] and in other high-risk fresh produce linked to outbreaks (i.e., basil, parsley, shredded carrots, and shredded cabbage with carrot mix) [34], in different types of berries [26], and in romaine lettuce [27] or complex matrices such as salsa/pico de gallo or guacamole [28]. The method was also validated for use in agricultural water using dead-end ultrafiltration filters (DEUFs) and small modifications in BMA Chapter 19b, published as BAM Chapter 19c [37]. In the present study, we proceeded with steps 2 and 3 of the BAM Chapter 19b method without modifications.

A limitation of the present study, as in any spiking experiment, is that there is always some inevitable inconsistency in the exact number of oocysts seeded per sample, due to pipetting variability, and minor variations in efficiency at each step of the procedure for each sample replicate are likely to contribute to small variations in the outcomes [19,34,38]. However, even with these variations, using the same oocysts and seeding procedure was useful for identifying significant differences in the present study of comparison of DNA extraction and detection methods, with the flotation procedure being more sensitive and able to detect small numbers of *C. cayetanensis* oocysts in soil samples using the BAM Chapter 19b qPCR as well as a new mitochondrial *C. cayetanensis* target qPCR.

## 5. Conclusions

In conclusion, the flotation procedure in the present study is an easy, reliable, highly sensitive, and low-cost method, which allowed for the detection of as few as 10 oocysts in 10 g of soil samples (1 oocyst/g). This method would be useful to identify environmental sources of *C. cayetanensis* contamination, as well as to investigate the role of soil in the transmission of *C. cayetanensis*, particularly in areas near portable toilets in agricultural produce farms. The effectiveness of any procedure for detection of pathogens, including parasites, in soil might depend on sample volume, soil texture, presence of organic matter, degree of soil contamination, among other factors. Further studies would need to be performed for different types of soil, and the procedure might require methodological adjustments for some of those soil types.

## Figures and Tables

**Figure 1 microorganisms-10-01431-f001:**
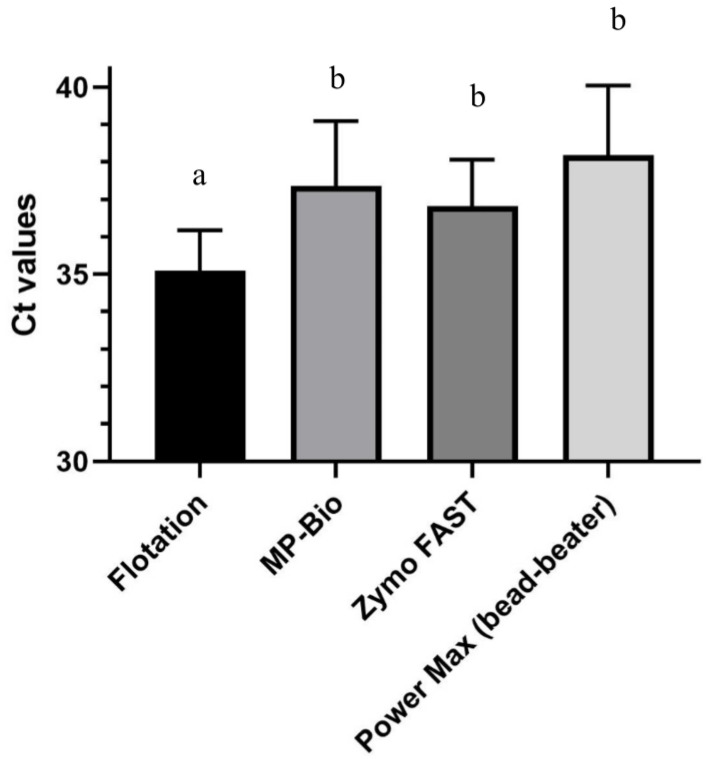
Comparison of average C_T_ values for *C. cayetanensis* detection in soil samples (10 g) seeded with 100 oocysts after DNA extraction by different methods (Flotation and BAM Chapter 19b Method 1; Fast DNA^TM^ 50 mL SPIN kit for soil (MP Biomedicals) Method 2, Quick-DNA ^TM^ Fecal/Soil Microbe Midiprep kit (Zymo Research), Method 3, and DNeasy^®^ PowerMax^®^ Soil Kit (Qiagen) using bead-beating, Method 4b). Different letters on top of methods indicate statistically significant differences (*p* < 0.05).

**Figure 2 microorganisms-10-01431-f002:**
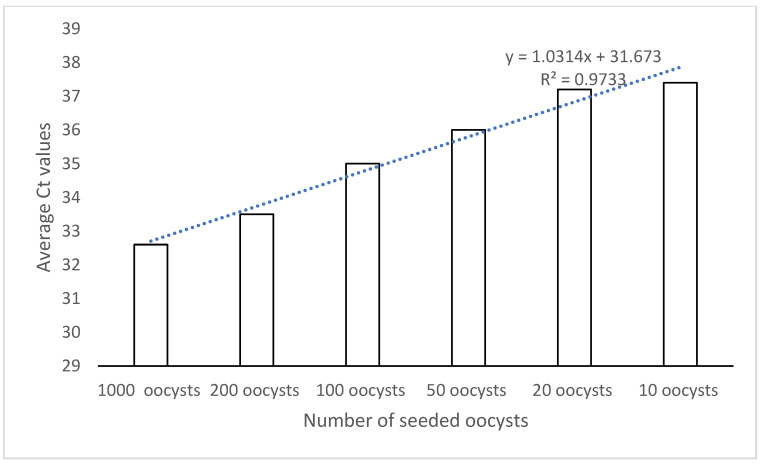
Linearity of *C. cayetanensis* detection of the flotation protocol in soil samples seeded with known numbers of oocysts.

**Table 1 microorganisms-10-01431-t001:** Characteristics of three commercial DNA isolation kits for soil and a concentration by flotation protocol following by DNA isolation as in BAM Chapter 19b, used in this study to extract genomic DNA from soil samples inoculated with 100 oocysts of *Cyclospora cayetanensis*.

	Flotation in Sucrose (1.12 s.d) and Fast DNA^TM^ SPIN Kit for Soil (MP Biomedicals) (Method 1)	Fast DNA^TM^ 50 mL SPIN Kit for Soil (MP Biomedicals) (Method 2)	Quick-DNA^TM^ Fecal/Soil Microbe Midiprep Kit (Zymo Research) (Method 3)	DNeasy^®^ PowerMax^®^ Soil Kit (Qiagen) (Method 4a—Vortexing)	DNeasy^®^ PowerMax^®^ Soil Kit (Qiagen) (Method 4b—Bead-Beating)
Size preps in kit	100	10	25	10	10
Bead beater: yes/no; Instrument recommended	Yes FastPrep^®^-24 instrument with provided adapter for Fastprep (24 × 2 mL tubes)	Yes FastPrep^®^-24 instrument and FastPrep^®^ BigPrepr Adapter (2 × 50 mL tubes)	Yes Bead beater (can be FastPrep^®^-24 instrument and FastPrep^®^ BigPrepr Adapter (2 × 50 mL tubes)	No Vortexing 10 min (vortex adaptor for 50 mL tubes (max 6 tubes). Alternatively, water bath set at 65 °C, shaking at maximum speed for 30 min	Yes FastPrep^®^-24 instrument and FastPrep^®^ BigPrepr Adapter (2 × 50 mL tubes)
User supplied reagents	100% ethanol	100% ethanol	Beta mercaptoethanol; 100% ethanol	None	None
Maximum soil sample	Performed after flotation (less than 0.4 g washed material)	Up to 10 g	5 g max (2.5 g recommended)	Up to 10 g	Up to 10 g
Column-based?	Yes	Yes	Yes	Yes	Yes
Steps in protocol	17	18	10	19	19 (substitution of vortexing in step 4 by bead-beating)
Incubation times	Increase DNA yield recommended by incubation at 55 °C for 5 min	No	No	Yes, 2–8 °C for 10 min (twice)	Yes, 2–8 °C for 10 min (twice)
Final elution volume	50–100 µL **	5 mL *	150 µL	5 mL *	5 mL *
Cost/reaction	(625/100) $6.25	(227/10) $22.7	(475/25) $19.0	(290/10) $29.0	(290/10) $29.0

* After DNA isolation in Method 2 and 4a and 4b, a DNA clean and concentrator-100 protocol step (Zymo Research, catalog No. D4030) was performed. Final elution volume after clean-up was 150 µL for comparison to the other methods. After DNA isolation from all protocols, quantitative real-time qPCR was performed following BAM Chapter 19b. ** Original elution volume was 75 µL. The total elution was diluted ½ to get a final 150 µL volume for comparison to the other methods.

**Table 2 microorganisms-10-01431-t002:** qPCR detection data (number of positive qPCR replicates, individual C_T_ values for *C. cayetanensis* and IAC for each sample) in samples of soil (10 g) seeded with 100 *C. cayetanensis* oocysts using different DNA isolation methods and molecular detection.

100 Oocysts	Flotation-Sucrose			Direct DNA Isolation-MPBio (Fastprep)			Direct DNA Isolation-Zymo Fast (Zymo)			Direct DNA Isolation-Power Max (Bead Beater)		
Sample Number	Number of Positive Reactions (Out of Three)	18 S C_T_ Value (Mean ± Standard Deviation	IAC C_T_ Value * (Mean ± Standard Deviation)	Number of Positive Reactions (Out of Three)	18 S C_T_ Value (Mean ± Standard Deviation	IAC C_T_ Value * (Mean ± Standard Deviation)	Number of Positive Reactions (Out of Three)	18 S C_T_ Value (Mean ± Standard Deviation	IAC C_T_ Value * (Mean ± Standard Deviation)	Number of Positive Reactions (Out of Three)	18 S C_T_ Value (Mean ± Standard Deviation	IAC C_T_ Value * (Mean ± Standard Deviation) **
1	3	34.6 ± 0.8	26.5 ± 0.4	0	Und	25.4 ± 0.1	3	36.6 ± 0.8	27.7 ± 0.6	3	35.6 ± 0.9	24.1 ± 0.2
2	3	33.9 ± 1.0	27.1 ± 0.1	3	36.3 ± 1.6	27.1 ± 0.2	3	35.8 ± 0.8	26.6 ± 0.2	2	36.7 ± 0.5	24.2 ± 0.0
3	3	35.2 ± 1.1	26.7 ± 0.4	2	36.4 ± 1.45	28.8 ± 0.4	3	36.1 ± 0.8	26.6 ± 0.5	1	38.0	26.4 ± 0.2
4	3	35.8 ± 0.1	26.4 ± 0.4	2	37.0 ± 1.3	26.4 ± 0.3	1	37.0	38.4 ± 2.3	1	36.7	25.2 ± 0.2
5	3	35.8 ± 1.2	26.1 ± 0.1	3	37.2 ± 1.0	33.8 ± 2.5	2	37.3 ± 1.0	30.5 ± 1.7	0	Und	26.8 ± 1.1
**Average**		**35.1 ± 0.8**	**26.6 ± 0.4**		**36.7 ± 0.4**	**28.3 ± 3.3**		**36.6 ± 0.7**	**30.6 ± 1.3**		**36.9 ± 1.5**	**25.4 ± 0.4**
Unseeded	0	Und	26.7 ± 0.2	0	Und	27.1 ± 0.5	0	Und	25.0 ± 0.1	0	Und	23.9 ± 0.1

* IAC of NTC in assay for Method 1: flotation = 26.1 ± 0.1; for direct DNA isolation soil in methods 2 and 3: 25.2 ± 0.2. ** IAC of NTC in Method 4b = 23.8 ± 0.1 Und: undetermined after 45 cycles (not detected). In red: Samples showing some degree of inhibition.

**Table 3 microorganisms-10-01431-t003:** qPCR detection data (number of positive qPCR replicates, individual C_T_ values) for *C. cayetanensis* detection in samples of soil (10 g) seeded with 20 and 10 *C. cayetanensis* oocysts using the flotation in dense sucrose solution method and molecular detection of the parasite.

Seeding Level	Flotation Method in Soil Samples (10 g)		
	Sample Number	Number of Positive Reactions (Out of Three)	18 S C_T_ Value (Mean ± Standard Deviation)
20 oocysts	1	2	37.9 ± 0.1
	2	3	36.9 ± 0.6
	3	3	37.7 ± 0.1
	4	2	37.9 ± 0.7
	5	1	36.5
	6	2	37.3 ± 1.9
	7	2	36.7 ± 1.3
	8	0	Und
	9	0	Und
	10	2	36.6 ± 0.0
10 oocysts	1	0	Und
	2	0	Und
	3	0	Und
	4	0	Und
	5	2	36.9 ± 0.1
	6	1	37.8
	7	2	37.6 ± 0.1
	8	0	Und
	9	0	Und
	10	0	Und

Und = Undetermined (not detected).

**Table 4 microorganisms-10-01431-t004:** qPCR Mit1C detection data (number of positive qPCR replicates, individual C_T_ values for *C. cayetanensis* and IAC for each sample) in samples of soil (10 g) seeded with 100 *C. cayetanensis* oocysts using different DNA isolation methods and molecular detection.

100 Oocysts	Flotation-Sucrose			Direct DNA Isolation-Power Max (Bead Beater)		
Sample Number	Number of Positive Reactions (Out of Three)	Mit1C C_T_ Value (Mean ± Standard Deviation)	IAC C_T_ Value * (Mean ± Standard Deviation)	Number of Positive Reactions (Out of Three)	Mit1C C_T_ Value (Mean ± Standard Deviation)	IAC C_T_ Value * (Mean ± Standard Deviation) **
1	3	32.0 ± 0.2	29.5 ± 0.6	3	34.2 ± 0.6	27.3 ± 0.1
2	3	31.5 ± 0.1	29.2 ± 0.2	3	34.7 ± 0.4	26.8 ± 0.1
3	3	31.5 ± 0.3	30.0 ± 0.8	3	33.3 ± 0.6	29.7 ± 0.0
4	3	32.4 ± 0.3	28.7 ± 0.1	3	36.1 ± 0.2	29.3 ± 0.1
5	3	33.1 ± 1.0	29.6 ± 0.3	3	34.0 ± 0.8	29.9 ± 0.2
**Average**		**32.1 ± 0.7**	**29.4 ± 0.5**		**34.5 ± 1.0**	**29.6 ± 0.3**
Unseeded	0	Und	29.9 ± 0.1	0	Und	28.6 ± 1.4

* IAC of NTC in assay for Method 1 (flotation) and Method 4b (commercial direct DNA isolation from soil) = 30.3 ± 0.1. ** Out of three replicates.

## Data Availability

Data are included in the manuscript.

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
