# Peer review of "Comparative Evaluation of an Easy Laboratory Method for the Concentration of Oocysts and Commercial DNA Isolation Kits for the Molecular Detection of Cyclospora cayetanensis in Silt Loam Soil Samples"

_microorganisms, 2022, doi:10.3390/microorganisms10071431_

Round 1

Reviewer 1 Report

The manuscript entitled Comparative Evaluation of an Easy Laboratory Method for the Concentration of Oocysts and Commercial DNA Isolation Kits for the Molecular Detection of Cyclospora cayetanensis in Silt Loam Soil Samples describes comparison study of three commercial DNA isolation kits with flotation method for identification Cyclospora cayetanensis in soil.
In my opinion the article is well written and provides interesting results for scientists from the field of parasitology and epidemiology as well. However, I have few issues for the authors which should be considered by them. Firstly, it’s a pity that the authors did not compare the isolation efficiency on samples at different levels of spiking e.g. 50 oocysts, 10 oocysts etc. It would be more beneficial  for the whole story. I would suggest doing such study and including this article or publishing another article with spiking samples. Otherwise, I would suggest changing the title and including “preliminary”.
I wonder if the authors have results from measuring the DNA concentration after isolation of each method and could they include such results in the article.

Author Response

Reviewer 1:

The manuscript entitled Comparative Evaluation of an Easy Laboratory Method for the Concentration of Oocysts and Commercial DNA Isolation Kits for the Molecular Detection of Cyclospora cayetanensis in Silt Loam Soil Samples describes comparison study of three commercial DNA isolation kits with flotation method for identification Cyclospora cayetanensis in soil.
In my opinion the article is well written and provides interesting results for scientists from the field of parasitology and epidemiology as well.

However, I have few issues for the authors which should be considered by them. Firstly, it’s a pity that the authors did not compare the isolation efficiency on samples at different levels of spiking e.g. 50 oocysts, 10 oocysts etc. It would be more beneficial for the whole story. I would suggest doing such study and including this article or publishing another article with spiking samples. Otherwise, I would suggest changing the title and including “preliminary”.

Thank you for your nice words.

After the comparison of the isolation/detection efficiency among the different methods using a seeding level (100 oocysts) and selecting the flotation method as the one with the best performance, we did different levels of spiking (1000, 200, 100, 50, 20 and 10 oocysts) and showed that the flotation method performed well in those conditions. We would not expect that further comparisons with different spiking levels would modify the conclusions of the present study.

I wonder if the authors have results from measuring the DNA concentration after isolation of each method and could they include such results in the article.

Although we do not usually measure the DNA concentration obtained in each sample, because based on our experience, DNA concentrations vary among samples, even when being processed by the same method, and the soil/produce DNA concentration is not representative of the parasite DNA present in the sample, following the reviewer advice, we measured the DNA concentration of representative soil samples processed in the study and we found variations among samples processed by the same method at different dates of processing (from 31.6 ng/ul to 62.8 ng/ul). Importantly, we observed that the quality of DNA from soil samples processed by flotation was higher at the A260/280 ratio (1.62-1.67) than those from the other methods compared in the study (1.41-1.48). We could add this information to the manuscript if the reviewer considers this information of interest.

Reviewer 2 Report

In this well-designed and reported study, the authors evaluate four methods of isolating DNA from Cyclospora cayetanensis in spiked soil samples. They tested the Qiagen DNeasy® PowerMax® Soil Kit with vortexing and bead beating, Zymo Quick-DNA TM Fecal/Soil Microbe Midiprep kit, MP Biomedicals Fast DNA 50 ml SPIN kit for soil, and flotation in sucrose (1.12 s.d) followed by the Fast DNA SPIN kit for soil. The sucrose floatation method consistently yielded amplifiable DNA for all of the replicates while the other methods failed for some replicates of each of the sample types. As expected in the results, the Ct values increased as the number of seeded oocytes was reduced and low seeded numbers of oocytes were prone to detection failure.

A lower input (0.25 g) Qiagen kit such as DNeasy® PowerSoil® Pro Kit with a lower elution volume (50-100 uL) might yield amplifiable and detectable DNA where the larger elution volume (5 mL) PowerMax and SPIN kit combo failed potentially due to soil humic material inhibition. The authors could test this option.

Specific edits:

Page 2, line 54: The reference author names should be deleted as they are indicated by number.  [1-(Ortega and Sanchez, 2010)]

Author Response

Reviewer 2

In this well-designed and reported study, the authors evaluate four methods of isolating DNA from Cyclospora cayetanensis in spiked soil samples. They tested the Qiagen DNeasy® PowerMax® Soil Kit with vortexing and bead beating, Zymo Quick-DNA TM Fecal/Soil Microbe Midiprep kit, MP Biomedicals Fast DNA 50 ml SPIN kit for soil, and flotation in sucrose (1.12 s.d) followed by the Fast DNA SPIN kit for soil. The sucrose floatation method consistently yielded amplifiable DNA for all of the replicates while the other methods failed for some replicates of each of the sample types. As expected in the results, the Ct values increased as the number of seeded oocytes was reduced and low seeded numbers of oocytes were prone to detection failure.

Thank you for the nice words. 

A lower input (0.25 g) Qiagen kit such as DNeasy® PowerSoil® Pro Kit with a lower elution volume (50-100 uL) might yield amplifiable and detectable DNA where the larger elution volume (5 mL) PowerMax and SPIN kit combo failed potentially due to soil humic material inhibition. The authors could test this option.

Thanks for the suggestion. Our main objective was to work with large samples of soil (5-10 g each), that would hopefully mean more possibilities to detect the parasite in field studies in soil. We would expect that working with 0.25 g samples will mean a lower sensitivity of detection working in field conditions. In the commercial kit in which elution was 5 ml, a clean and concentration kit was used to eliminate as much as possible soil inhibitors and concentrate the DNA to the same volume as the other kits for comparison. 

Specific edits:

Page 2, line 54: The reference author names should be deleted as they are indicated by number.  [1-(Ortega and Sanchez, 2010)]. Deleted.